# BENTO: BENCHMARK TASK REDUCTION WITH IN-CONTEXT TRANSFERABILITY

**Hongyu Zhao[1], Ming Li[1], Lichao Sun[2], Tianyi Zhou[1]**
[1]University of Maryland, College Park
[2]Lehigh University
{hongyuz, minglii, tianyi}@umd.edu
Project: https://github.com/tianyi-lab/bento

## ABSTRACT

Evaluating large language models (LLMs) is costly: it requires the generation and examination of LLM outputs on a large-scale benchmark of various tasks. This paper investigates how to efficiently reduce the tasks used to benchmark LLMs without affecting the evaluation quality. Our study reveals that task transferability and relevance provide critical information to identify the most representative subset of tasks via optimizing a facility location function. We propose a practically efficient metric for estimating the transferability between two tasks via in-context learning (ICL). By analyzing the pairwise transferability, we can reduce tasks in a modern LLM benchmark (e.g., MMLU or FLAN) to 5% while inducing only a $< 4\%$ difference to the evaluation on the original benchmark. Compared to prior works, our method is training-free, gradient-free, and highly efficient requiring ICL only.

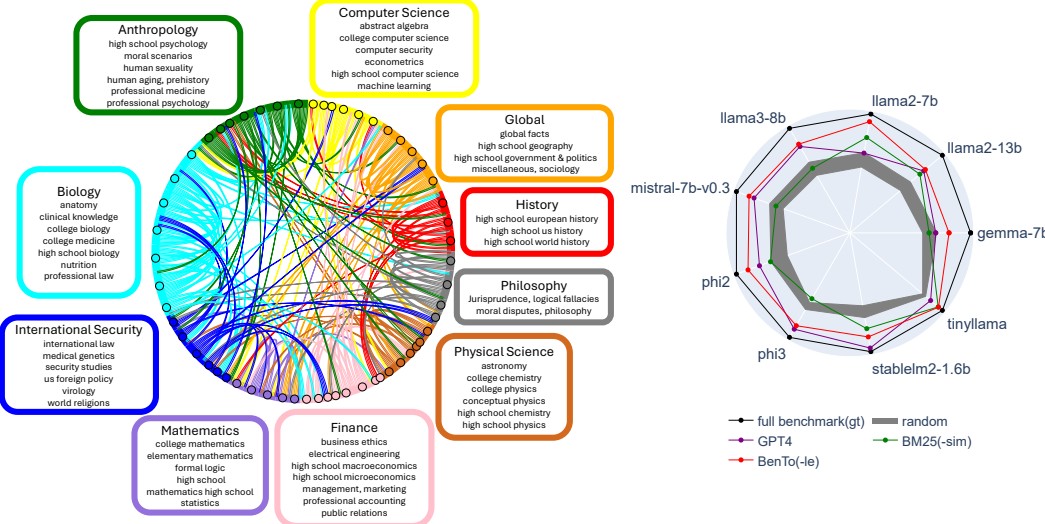

Figure 1: **LEFT: In-context Transferability (ICT) reveals the clusters of benchmark tasks.** We apply spectral clustering to ICT (arcs[1]) between MMLU tasks (nodes), whose color denotes the cluster it belongs to. The discovered clusters are associated with explainable themes. The theme and tasks of each cluster are listed around the chord graph. Only the top-7% arcs with the highest ICT values are shown in the graph, among which intra-cluster arcs are much more than inter-cluster arcs, implying a "sparse" topology captured by ICT. **RIGHT: Evaluation accuracy of task reduction methods**. Each method selects 3 out of the 57 tasks in MMLU to evaluate 9 LLMs (axes). The plot reports $1 - |\sigma - \sigma^*|/\sigma^*$ in log-scale where $\sigma$ and $\sigma^*$ are the evaluation metrics on the reduced-benchmark and full-benchmark, respectively. Our method (BENTO-le) achieves 97% evaluation accuracy on average. The grey band reports the random selection baseline's mean±standard variation. All baselines are defined in Section 5. Table 2 reports the result when selecting different number of tasks.

---

[1]Each arc connects a source task with a target task and has the same color as the source task.

## 1 INTRODUCTION

Evaluation of large language models (LLMs) is critical to examining the versatile capability and identifying possible weaknesses/risks of LLMs before deploying them to downstream tasks in practice. However, the development of the LLM benchmark is still an open challenge (Chang et al., 2023; McIntosh et al., 2024): it is usually expensive and heavily relies on human involvement, yet it is still unclear how large the benchmark should be to deliver reliable and consistent evaluation results. In practice, to cover various different application scenarios and diverse skill sets of LLMs, current LLM benchmarks usually attempt to include sufficient test cases drawn from as many tasks as possible (Hendrycks et al., 2021a;b; Wei et al., 2021), e.g., tens to hundreds. Due to the expensive sequential decoding of autoregressive LLMs, larger benchmarks greatly increase the evaluation cost and lead to severe overhead in the development process of LLMs.

The substantial costs associated with LLMs drive the need to explore the feasibility of reducing the number of tasks in LLM benchmarks without compromising their evaluative capabilities. Our study in this paper investigates the transferability (Vu et al., 2020; Jiang et al., 2022) between benchmark tasks to discern their relevance and potential overlap. Transferability indicates that skills or knowledge acquired in one task (task-$i$) can significantly enhance performance in another (task-$j$). Therefore, a model demonstrating strong performance on task-$i$ is likely to perform well on task-$j$, leveraging the inherent generalization capabilities of LLMs. Given an accurate estimation of the task transferability, we may reduce the tasks required in LLM benchmarking and thus optimize the evaluation efficiency.

Existing transferability estimation (Nguyen et al., 2020; Bao et al., 2019; Tan et al., 2021) mainly rely on model finetuning or Fisher information, which is computationally prohibitive for LLMs considering the total number of benchmark tasks and large model size. Hence, this paper aims to design a cost-efficient and training-free approach to measure the transferability between different tasks. Motivated by the current progress of in-context learning (ICL) (Brown et al., 2020; Dong et al., 2023), we propose **in-context transferability (ICT)** as a training-free approach tailored for benchmark reduction. Specifically, when applying task-$i$'s exemplars as the context for task-$j$'s queries, it provides an effective low-cost estimation of the transferability from task-$i$ to task-$j$. The resulting improvement over task-$j$'s zero-shot (non-context) performance reflects the merits of task-$i$'s knowledge of task-$j$.

Thorough analysis is conducted towards the transferability matrix computed on all the pairs of tasks from MMLU (Hendrycks et al., 2021a;b), a widely adopted LLM benchmark. In the visual representation provided by Figure 1 (LEFT), we observe a 'sparse' clustering pattern. This pattern is characterized by a concentration of dense interconnections among tasks around the periphery of the circle, with noticeably fewer connections traversing the central area, indicating that the intra-cluster transferability is larger than the inter-cluster transferability. This observation leads to a Laplacian Eigenmaps (LE) (Belkin & Niyogi, 2003) embedding of tasks, which is also known as the first step of spectral clustering.

To effectively extract a representative subset of tasks that mirrors the full scope of the original benchmark, we propose **Ben**chmark **T**ask reducti**O**n (**BENTO**) that formulates the task selection into a facility location (FL) problem (Cornuejols et al., 1977). In BENTO, the task similarities are derived either directly from the similarity matrix computed via Laplacian Eigenmaps (LE) or are recalculated within the LE-embedded space. The FL objective was to maximize the similarity between each task in the benchmark and the closest task in the reduced subset. This objective is submodular, allowing us to employ a greedy algorithm (Nemhauser et al., 1978) that efficiently achieves a high-quality approximate solution. Extensive experiments are conducted to evaluate the effectiveness of BENTO-reduced benchmark by comparing the performance of several widely used LLMs on both the reduced and original benchmarks. Remarkably, as is shown in Figure 1 (RIGHT) and Table 1, the results are highly consistent, even though the reduced benchmark comprises only 5% of the original tasks. This finding underscores the efficiency of our approach. When compared to existing benchmark reduction methods, BENTO not only yields more accurate evaluation results but also significantly lowers the costs associated with transferability estimation. Furthermore, ICT offers substantial potential benefits across a wide array of LLM research problems and applications, making it a topic of independent interest.

Our main contributions can be summarized as:

Table 1: Comparison of different benchmark reduction methods on MMLU. The metric is the difference between performance on the full/original benchmark and the reduced benchmark. Previously reported results are available in Appendix G[2].

| Model | Full Benchmark(100%) | BENTO(5%) | CHATGPT(5%) | BM25(5%) |
|---|---|---|---|---|
| Llama2-13b | 54.5 | **-0.6** | +5.4 | -6.8 |
| Llama2-7b | 46.0 | **+3.8** | +6.8 | -4.6 |
| Llama2-70b | 66.6 | **+1.3** | +6.2 | -9.0 |
| Llama3-8b | 61.7 | **-1.5** | +4.2 | -11.6 |
| Mistral-7b-v0.3 | 62.1 | **+0.1** | +4.9 | -10.6 |
| Phi-2 | 56.5 | **+0.2** | +5.1 | -8.2 |
| Phi-3-mini-4k | 69.5 | **+0.5** | +2.1 | -12.2 |
| StableLM-2-1.6B | 34.6 | **+0.1** | -1.6 | -3.5 |
| TinyLlama | 24.9 | +1.0 | +1.4 | **-0.4** |
| Gemma-7b | 65.2 | **-1.8** | +9.0 | -10.8 |
| QWEN2.5-7b | 74.2 | **+1.6** | +4.2 | -14.7 |
| QWEN2.5-14b | 79.8 | **+1.8** | +3.5 | -14.0 |
| GPT-4o-mini | 74.3 | **+0.8** | +5.0 | -11.6 |
| GPT-4o | 68.9 | **+0.9** | -2.2 | -8.8 |

- **In-Context Transferability (ICT):** We harness in-context learning to estimate task transferability and discover graph and clustering structures of benchmark tasks. ICT does not require any finetuning and provides the first scalable transferability metric on LLMs.
- **Benchmark Task Reduction (BENTO):** We develop an efficient benchmark reduction approach, which selects a representative subset of tasks according to ICT. It can reduce tasks in an LLM benchmark to only 5%, which substantially reduces LLM evaluation costs without hurting the quality.

## 2 RELATED WORK

**Task Transferability.** Efficient estimation of task transferability has been a long-studied problem. Past works mainly use Bayesian optimization (Weiss et al., 2016) and information theory (Bao et al., 2019; Tan et al., 2021). LEEP (Nguyen et al., 2020) proposes estimating via approximately training the model by linear probing on the data of the source tasks and evaluating the target tasks, which resembles our in-context learning approach. Xia et al. (2024) leverages similarity between gradient features obtained by training with LoRA (Hu et al., 2022) as a transferability measure, which inspires us to transform the performance feature into similarity matrices.

**Benchmark Reduction.** Dataset reduction for LLM training Li et al. (2024b;a) has been a heated area while the reduction for benchmarks is till under-explored. Current benchmark reduction methods can be categorized into two major approaches: selecting tasks from a benchmark and selecting examples from a single task. Our work falls into the first one. It may seem that the second approach is more robust at least on non-few-shot benchmarks (Perlitz et al., 2024), but we can prove that combining the two approaches always yields a better result (see Section 5.2), so these two approaches parallel and we can apply them one by one. Ye et al. (2023) also goes in the first direction and analyzes Big-bench (bench authors, 2023). However, it's time-consuming to collect the training data they need, as their methods require performances from different models with various parameters as guidance. Polo et al. (2024) goes in the second direction, making use of item response theory (IRT) models, but share the same drawback. On the contrary, our method only requires the performance of a single model as guidance, making it very efficient in data collection. Vivek et al. (2024) is a relatively efficient example-selection method, which, similar to us, also draws insight from clustering. Yet, their approach can only predict the ranking instead of the exact performance of models on the whole benchmark.

---

[2]They may use slightly different prompts and distinct random seeds, which are not released.

# 3 TASK TRANSFERABILITY ANALYSIS BY IN-CONTEXT LEARNING

Aiming at proposing a cost-efficient benchmark reduction approach, we first analyze the structure of benchmarking tasks by studying the transferability between each pair of tasks. Given source task-$i$ and target task-$j$, the transferability from task-$i$ to task-$j$ is often measured by training a model on task-$i$ and then evaluating it on task-$j$. However, this approach requires training per task and thus is not computationally scalable to modern LLMs and many tasks. We propose to harness in-context learning to provide a training-free estimation of the transferability between tasks.

**Compute in-context transferability (ICT) embedding matrix** $A$. To estimate the transferability from source task-$i$ to target task-$j$, we randomly sample $L$ exemplars $e_{1:L}^{(i)}$ from task-$i$, where each $e_k^{(i)} \triangleq (x_k^{(i)}, y_k^{(i)})$ is an input-output pair for task-$i$. They are combined with the instruction of task-$i$ $p^{(i)}$ to constitute the context, which is used to query an LLM's answer to each question $x^{(j)}$ of task-$j$, i.e., $\text{LLM}([p^{(i)}, e_{1:L}^{(i)}, x^{(j)}])$. The performance of such *transfer ICL* reflects the transferability from task-$i$ to task-$j$: If task-$j$ shares a more similar format, theme/topics, or can benefit more from the knowledge of task-$i$, then it's more likely that the transfer ICL can improve the performance on task-$j$. To reduce the variance, we can repeat the transfer ICL $M$ times with different random seeds and estimate the transferability by their average.

To study the structure of a multi-task benchmark, we estimate a matrix of task transferability for all the pairs of tasks by applying the above operation to each pair. Assuming that we have $N$ tasks in total, then we will get an $N \times N$ transferability matrix $A$, where $A_{ij}$ is an estimation of the *in-context transferability (ICT)* from task-$i$ to task-$j$, i.e.,

$$A_{ij} = \frac{1}{n_j} \sum_{k=1}^{n_j} s\left(\text{LLM}([p^{(i)}, e_{1:L}^{(i)}, x_k^{(j)}]), y_k^{(j)}\right),\tag{1}$$

where $n_j$ is the total number of input-output pairs we sampled from task-$j$. $s(\cdot, \cdot)$ is an evaluation metric such as an exact match or similarity score. A larger $s(\cdot, \cdot)$ indicates a better transfer-ICL's performance. To further reduce variance, we resample the $n_j$ questions multiple times using different random seeds and average the achieved $A_{ij}$. Since target tasks may differ in difficulty, we normalize $A$ by zero-centering each column of $A$, i.e.,

$$A_{ij} \leftarrow A_{ij} - \frac{1}{N} \sum_{i=1}^{N} A_{ij}.\tag{2}$$

In the normalized $A$, each row can be viewed as an embedding vector for the corresponding task.

**Spectral clustering.** We investigate the graph structure among tasks by applying clustering based on the tasks' feature matrix $A$, which can reveal whether the transferability between intra-cluster tasks is high (and thus they can be further reduced). Since $A$ defines the pairwise transferability on a graph of tasks, we choose to use spectral clustering, a widely used algorithm for graph cut problems. Given $A$, spectral clustering computes a similarity matrix that is symmetric and non-negative, by applying a Euclidean similarity kernel to $A$:

$$S_{ij} = c - E_{ij}, \ E_{ij} = \sqrt{\sum_{k=1}^{N} (A_{ik} - A_{jk})^2}\tag{3}$$

where $c$ is a constant to ensure the non-negativeness of $S$. We discuss the choice of $c$ in Section 5.2. With a similarity matrix $S$ given, spectral clustering can be performed as shown in Figure 1, where each cluster is defined by a very clear theme. For example, the red cluster contains all three tasks about history in the MMLU benchmark. The clustering result aligns well with human intuitions, which demonstrates the effectiveness of ICT on capturing the inter-task transferability and redundancy between benchmarking tasks. Note that there are a few tasks with counter-intuitive

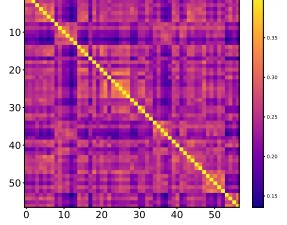

(a) $S$, induced by ICL embedding $A$.

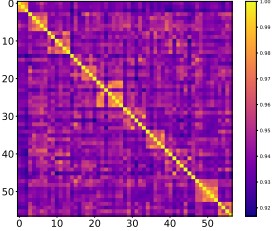

(b) $S'$, induced by LE embedding $A'$.

Figure 2: Similarity matrices.

clustering assignments, e.g., "professional law" is in the biology cluster. This may indicate that our ICT representation captures information inherent to the task structures, which cannot be inferred solely from their names. We will come back to this in Section 5.

The arcs in the chord graph indicate which tasks are more frequently good source tasks. As we've mentioned in Section 1, this clustering has an interesting structure: The arcs within each cluster are more than those between clusters. This might seem trivial at first by definition of spectral clustering, but note that the arcs come from the ICT feature matrix $A$, which is not the feature matrix that we directly perform clustering algorithms on.

**LE embedding.** Let's take a closer look at the process of spectral clustering. We view the similarity matrix $S$ as the adjacency matrix of a complete graph. We first compute

$$
\begin{aligned}
A' &= \text{Eigenvector}_{1:K}(L), \\
L &= I - D^{-\frac{1}{2}} S D^{-\frac{1}{2}}, \quad D = \text{Diag}(S\vec{1}).
\end{aligned}
\tag{4}
$$

where $K$ is a hyperparameter. We then perform the K-Means clustering to the rows of $A'$. As expected, Figure 2 shows that the similarity matrix $S'$ induced by $A'$ (Figure 2b) indeed has the same clustering structure as $S$ but exhibits a stronger block-diagonal pattern than $S$ induced by $A$ (Figure 2a). This inspires us to view $A'$ as an alternative task embedding of $A$: It only preserves and strengthens neighborhood information of $A$, thus less noisy compared to the original features (Belkin & Niyogi, 2003). The process to transform $S$ into $A'$ is called Laplacian Eigenmaps (LE), so we call $A'$ LE embedding in this paper.

The clustering results motivate us to select tasks that are more important or representative: these tasks exhibit higher transferability to more tasks than others, and each of them may transfer to a different subset of tasks. If we can select a subset of representative tasks that can cover most of the tasks in a benchmark, then we can use the performance on this subset to approximate the performance on the whole benchmark! That is the benchmark-task reduction problem we'll discuss in the next section.

## 4 BENCHMARK-TASK REDUCTION (BENTO) BY FACILITY LOCATION

Benchmark-task reduction (BENTO) aims to select a fixed-size subset of tasks, such that the performance of any model on this subset serves as a proxy of the model's performance on the whole benchmark. Intuitively, we want to choose the most "representative" subsets of tasks. In the context of transferability, the representativity of a subset of tasks can be measured by their transferability to other tasks in the benchmark. If a subset can transfer to the whole benchmark well, then the performance on this subset will be highly correlated to the performance on the whole benchmark. Under the assumption that the task difficulty in a benchmark is on similar levels, the model performance on this subset can directly serve as a prediction of the performance on the whole benchmark, and this holds for any model.

How do we choose such a subset? For every target task in the benchmark, if we can always find a source task in our subset with sufficiently high transferability to the target, then the subset can transfer to all the tasks in the benchmark. On the other hand, if there exist two or more tasks with high transferability to the same group of tasks, then retaining only one of them suffices to keep the transferability of the representative subset. Inspired by these intuitions, we aim to find a subset such that the similarity between each task in the benchmark and its "nearest" task in the subset is maximized. This formally reduces to optimizing the facility location (FL) function, i.e.,

$$
X^* \in \arg \max_{X \subseteq 2^N, |X| \leq k} f(X) \triangleq \sum_{i=1}^{N} \max_{j \in X} S_{ij}.
\tag{5}
$$

A larger $f(X)$ corresponds to a more representative subset of tasks $X$. Since this function is submodular, the optimization problem can be solved explicitly and efficiently with a greedy algorithm.

As an alternate to $S$ in Equation (3) (which might be affected by the long-range noises in $A$), we derive a cosine similarity matrix $S'$ from the LE embedding $A'$:

$$
S'_{i,j} = \frac{\langle A'_i, A'_j \rangle}{\|A'_i\| \cdot \|A'_j\|}.
\tag{6}
$$

As shown in Figure 2, $S'$ shares a lot of common properties with $S$, so we propose a variant that replaces $S$ in Equation (5) with $S'$. The LE embedding removes some long-range noise and is expected to be more robust in realistic scenarios. We call the original version using $S$ "BENTO-sim" and the LE variant using $S'$ "BENTO-le", where BENTO is an abbreviation of **Ben**chmark **T**ask reducti**O**n.

A detailed algorithm is presented in Appendix A. To summarize, our pipeline is: first compute a transferability matrix via in-context learning, then compute a similarity matrix based on the transferability matrix, and maximize the facility location function defined by the similarity matrix to select a subset of representative tasks.

# 5 EXPERIMENT

**Benchmarks.** We assess our method mainly on two benchmarks: MMLU (Hendrycks et al., 2021a;b) and FLAN (Wei et al., 2021). MMLU is a question-answering dataset containing 57 tasks with diverse subjects and levels of difficulty, mainly focusing on the knowledge of the language models. All the questions in MMLU are multiple-choice questions with 4 options for each answer. On MMLU, we use accuracy (ACC) as the evaluation metric $s(\cdot, \cdot)$ in Equation (1). FLAN is a dataset with more diverse forms of tasks, including free-form generation tasks like translation and summarization. Since FLAN is a large dataset, we sampled 100 questions from each of its 66 tasks as our "whole benchmark". On FLAN, we use response perplexity as the evaluation metric $s(\cdot, \cdot)$. We follow widely used prompts for these benchmarks without any re-engineering, more details in Appendix D.

**Evaluation Metric.** In both datasets, we apply all methods to select $k$ representative tasks with $k$ from 1 to $K$ (We pick $K$ to be approximately 18% of the tasks, i.e., 10 on MMLU and 12 on FLAN). For each value of $k$, we calculate the root mean square error (RMSE) of the predicted performance (i.e. performance on the selected tasks) across all the models. To ensure comparability, the RMSE is normalized by the root mean square (RMS) of the ground truth performance, yielding the following normalized RMSE:

$$\text{NRMSE} = \sqrt{\frac{\sum_{t=1}^{T}(\sigma_t - \sigma_t^*)^2}{\sum_{t=1}^{T}\sigma_t^*}}, \quad (7)$$

where $T$ denotes the total number of evaluated models, $\sigma_t$ and $\sigma_t^*$ denote the evaluation metrics of model $t$ on the reduced-benchmark and the original full-benchmark, respectively. The NRMSE measures the relative error (error rate) of the reduced benchmark in terms of $L_2$ error. For a more fine-grained evaluation of each model, please refer to Table 1 and Appendix E.

**Models.** ICL is performed on Llama-2-13B (Touvron et al., 2023) and Llama-2-7B to estimate ICT for MMLU tasks and FLAN tasks, respectively. Since the goal is to find a reduced benchmark that can replace the original benchmark to evaluate any models, we compare the benchmark evaluation results on 14 popular LLMs, including Llama-2-70B,Llama-2-13B, Llama-2-7B, Llama-3-8B (Dubey et al., 2024), Gemma-7B (Team et al., 2024), Phi-2 (Javaheripi et al., 2023), Phi-3 (Abdin et al., 2024), StableLM-2-1.6B (Bellagente et al., 2024), Mistral-7b-v0.3 (Jiang et al., 2023), TinyLlama (Zhang et al., 2024), QWEN2.5-7B, QWEN2.5-14B (Yang et al., 2024), GPT-4o-mini and GPT-4o (Hurst et al., 2024).

**Baselines.** We compare BENTO with the following baselines:

- **random**: A simple baseline involves randomly selecting tasks. To reduce variance, we sample 1000 sets of $k$ random tasks for each $k$ and compute the average NRMSE across these samples.

- **GPT4** (OpenAI, 2023): We prompt GPT4 to suggest representative tasks and rank them based solely on the names of the tasks. Given that our evaluation is based on well-established benchmarks, GPT-4 likely has prior knowledge of these tasks and may have encountered them during training.

- **BM25-sim**: BM25 (Robertson et al., 2009) is a classic measure of text similarity. Here, we calculate the BM25 score between each task's corpses (including instructions, solutions, etc.) and use it to replace the ICL transferability matrix $A$. The remaining steps are the same as BENTO-sim.

- **BM25-le**: A variant of **BM25-sim**, which uses $S'$ for the FL problem, just as in **BENTO-le**.

Table 2: NRMSE on MMLU (lower the better) when selecting $k$ tasks for evaluation. Each number is averaged over different models. The standard deviation can be found in Appendix C.

| Method | Best | k=1 | k=2 | k=3 | k=4 | k=5 | k=6 | k=7 | k=8 | k=9 | k=10 |
|---|---|---|---|---|---|---|---|---|---|---|---|
| random | 0.091 | 0.228 | 0.163 | 0.144 | 0.130 | 0.118 | 0.110 | 0.104 | 0.100 | 0.095 | 0.091 |
| GPT4 | 0.034 | 0.213 | 0.078 | 0.093 | 0.077 | 0.056 | 0.034 | 0.160 | 0.180 | 0.168 | 0.135 |
| BM25-le | 0.074 | 0.074 | 0.218 | 0.217 | 0.216 | 0.205 | 0.193 | 0.199 | 0.213 | 0.209 | 0.176 |
| BM25-sim | 0.093 | 0.351 | 0.188 | 0.177 | 0.119 | 0.101 | 0.093 | 0.162 | 0.137 | 0.134 | 0.117 |
| BENTO-le | 0.029 | **0.051** | 0.098 | **0.029** | **0.049** | **0.045** | **0.031** | **0.060** | 0.168 | 0.147 | 0.113 |
| BENTO-sim | **0.026** | 0.123 | **0.061** | 0.177 | 0.119 | 0.101 | 0.039 | 0.079 | **0.039** | **0.029** | **0.026** |

## 5.1 MAIN RESULTS

Results on the MMLU benchmark are presented in Table 2. As shown, the best of our two methods consistently outperforms other approaches. Notably, Our method can achieve an error rate of 3% with only 3 tasks out of 57, which constitutes approximately 5% of the total number of tasks (5.8% of test samples), significantly surpassing the baseline methods. This demonstrates that the information embedded in the ICL transferability matrix can be effectively utilized for benchmark reduction.

A deeper examination of the performance of the random and GPT-4 baselines reveals some intriguing patterns. Expectedly, the NRMSE of the random baseline always decreases as $k$ increases. In contrast, while the GPT-4 baseline exhibits a general downward trend in NRMSE as $k$ increases, an anomalous spike occurs at $k = 7$. Upon closer inspection, GPT-4 selects the task "professional law" as the seventh task, justifying its choice with the reasoning that this task is "relevant for understanding societal structures." This decision seems intuitively sound from a human perspective, as professional law often deals with governance and social order. However, our clustering result in Figure 1 suggests otherwise: the task "professional law" is actually placed in the biology cluster, revealing an unexpected underlying connection. This discrepancy highlights a key advantage of our approach. The fact that our methods consistently outperform the GPT-4 baseline indicates that our task representations capture more nuanced and accurate relationships between tasks beyond what their names or surface-level associations suggest.

Table 3: NRMSE on FLAN (lower the better). $k$ is the number of selected tasks. Each number is averaged over different models. The standard deviation can be found in Appendix C.

| Method | Best | k=1 | k=2 | k=3 | k=4 | k=5 | k=6 | k=7 | k=8 | k=9 | k=10 | k=11 | k=12 |
|---|---|---|---|---|---|---|---|---|---|---|---|---|---|
| random | 0.49 | 1.27 | 1.06 | 0.92 | 0.83 | 0.77 | 0.70 | 0.64 | 0.60 | 0.56 | 0.53 | 0.51 | 0.49 |
| GPT4 | 0.09 | 7.48 | 3.29 | 1.86 | 1.36 | 0.89 | 0.58 | 0.61 | 0.41 | **0.26** | **0.13** | **0.09** | 0.44 |
| BM25-le | 0.51 | 0.99 | 0.76 | 0.61 | 0.57 | 0.58 | 0.65 | 0.51 | 1.61 | 1.33 | 1.10 | 1.36 | 1.36 |
| BM25-sim | 0.24 | 7.93 | 4.36 | 2.57 | 1.69 | 1.30 | 0.93 | 0.65 | 0.68 | 0.49 | 0.36 | 0.31 | 0.24 |
| BENTO-le | 0.07 | **0.55** | **0.47** | 0.22 | **0.07** | 1.44 | 1.03 | 0.86 | 0.63 | 0.58 | 0.50 | 0.40 | 0.29 |
| BENTO-sim | **0.04** | 0.93 | 0.76 | **0.04** | 0.10 | **0.17** | **0.30** | **0.26** | **0.25** | 0.33 | 0.35 | 0.18 | **0.21** |

Our main results on FLAN are shown in Table 3. On FLAN, our method achieves an error rate of 4% using approximately 5% of the total number of tasks. Note that here the GPT4 baseline performs relatively better compared to its performance on MMLU. This improvement is primarily because the task names in FLAN are more informative —— they are names of well-known, established datasets such as SST-2. GPT-4 has likely encountered these tasks during its training, enabling it to infer the content of each task without needing to analyze individual examples. Despite this, our methods still outperform it for most values of $k$. Our method's consistent performance across different benchmarks indicates its potential applicability to a wide range of tasks and datasets.

The results of the similarity-based methods on both the MMLU and FLAN datasets exhibit a consistent pattern: BENTO-le performs well when the value of $k$ is sufficiently small, whereas BENTO-sim demonstrates better performance for larger values of $k$. This pattern also applies to the BM25 baseline; initially, BM25-sim underperforms compared to BM25-le but surpasses it as $k$ increases. This trend is clearly illustrated in Figure 3, where we plot the difference ($\Delta$) of performance of "sim" methods and "le" methods. The underlying reason for this behavior lies in the properties of

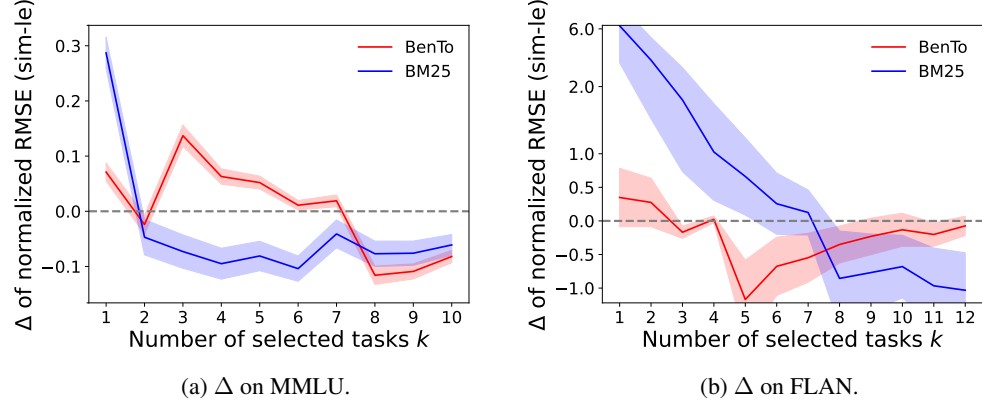

(a) $\Delta$ on MMLU.              (b) $\Delta$ on FLAN.

Figure 3: Difference ($\Delta$) in NRMSE between $S$ ("sim") and $S'$ ("le") when used to select different numbers of tasks (x-axis). Larger $\Delta$ indicates the "le" variant produces a better reduced benchmark than "sim". **For both BENTO and BM25, "le" is better ($\Delta \geq 0$) for smaller $k$ while "sim" is better ($\Delta \leq 0$) for larger $k$.**

Laplacian Eigenmaps. As previously discussed, Laplacian Eigenmaps are designed to preserve local neighborhood relationships while discarding long-range information. This characteristic makes them highly effective when selecting a small number of tasks, where local similarities are paramount. However, as the number of selected tasks increases, the importance of capturing global structure and long-range relationships becomes more significant. Consequently, methods that consider long-range similarities become more effective for larger $k$ values.

Experiments on other benchmarks (BBH (Suzgun et al., 2022) and AgiEval (Zhong et al., 2023)) can be found in Appendix F.

## 5.2 ABLATION STUDY

Table 4: Ablation study of similarity metrics: we compare the best NRMSE on different datasets achieved by different metrics: "cos"– cosine similarity, and "cheby"– Chebyshev similarity.

| Method | MMLU | AGIEval Eng | Big Bench Hard |
|---|---|---|---|
| cheby-le | 0.05 | 0.07 | 0.09 |
| cheby-sim | 0.11 | 0.06 | 0.22 |
| cos-le | 0.05 | **0.03** | 0.20 |
| cos-sim | 0.03 | 0.09 | 0.10 |
| BENTO-le | **0.03** | **0.03** | **0.05** |
| BENTO-sim | **0.03** | **0.03** | 0.15 |

**Choice of $c$.** When calculating the similarity matrix $S$ from the ICL transferability $A$ using Equation (3), we need to choose a hyperparameter $c$. Note that by definition $c$ does not influence the results of BENTO-sim; however, it does impact the performance of BENTO-le. In our main experiments, we set $c = 1.5 \max_{i,j}(E_{ij})$, ensuring that $S \geq 0.5 \max_{i,j}(E_{ij}) > 0$. This specific choice was empirically validated on the MMLU dataset, where it produced reasonable clustering results. But how optimal is this choice? Could other values of $c$ yield better performance? Would this selection generalize effectively to FLAN?

To address these questions, we parameterized $c$ as $c = t \max_{i,j}(E_{ij})$, where $t$ is sampled from a uniform distribution $U(1, 50)$, and generated 1000 random values of $t$. We then evaluated the performance of BENTO-le under these different values of $c$. On MMLU, out of the 1000 samples, 191 led to better average performance over $k$ compared to our original choice, while 340 achieved better performance on the best $k$. In contrast, on the FLAN dataset, 506 samples improved the average performance, and 872 samples enhanced the best performance compared to our initial selection of $c$.

These findings suggest that while our choice of $c$ is reasonably effective on MMLU, it is less optimal on FLAN. This variability in performance across datasets indicates that a more adaptive approach to selecting $c$ could be beneficial. In future work, we could explore dynamic strategies for setting $c$, potentially based on specific dataset characteristics or performance metrics. This approach could lead to more consistent improvements across different benchmarks.

**Choice of similarity metrics.** In our study, we opted for Euclidean similarity when computing the similarity matrix $S$. An important question arises: how would other similarity metrics, such as cosine similarity or Chebyshev similarity, affect the results? As shown in Table 4, while other metrics like cosine and Chebyshev similarity produced results that were slightly worse than Euclidean similarity, the performance gap was not large, especially between cosine similarity and Euclidean similarity. This suggests that Euclidean similarity may offer a slight advantage on the specific benchmarks we evaluate on, but other similarity measures could still be viable alternatives on different benchmarks.

**Task selection and example selection.** While one might argue that selecting representative examples within each task could yield better results while keeping inference costs low, it is important to note that our method can be combined with existing example-selection techniques to further enhance the reduction rate. To evaluate this, we compare two approaches: randomly selecting examples from each task with and without incorporating BENTO. The results, reported in Table 5, demonstrates that BENTO can further reduce the NRMSE of the 5.0% example-selection baseline ("Random") by reducing the examples to 2.0% using task selection.

When the number of examples per task becomes extremely limited, continuing to reduce examples per task leads to a substantial increase in NRMSE, suggesting that task selection offers a more robust alternative in such scenarios. Therefore, task selection and example selection are complementary strategies that can be effectively combined to achieve higher reduction rates.

Table 5: Example selection with and without BENTO (-sim) on MMLU. "Random" refers to random selection of examples. "Random+BENTO" applies "Random" at first to reduce the examples per task to 5% and then selects a subset of tasks by BENTO. It shows that **BENTO can further improve example selection and outperforms example selection only**. For example, "Random+BENTO" with 2.0% remaining data achieves a lower NRMSE than "Random" with 5.0% remaining data; "Random+BENTO" with 0.7% remaining data achieves the same NRMSE as "Random" with 2.0% remaining data.

| Selection strategy | # Selected Tasks | Remaining Examples (%) | NRMSE |
|---|---|---|---|
| Random | 57 | 5.0% | 0.029 |
| Random | 57 | 0.7% | 0.109 |
| Random + BENTO | 11 | 0.7% | **0.051** |
| Random | 57 | 2.0% | 0.051 |
| Random + BENTO | 21 | 2.0% | **0.026** |

**Facility location v.s. K-medoids clustering.** We select FL for its efficiency and precise formation of the task reduction problem. To evaluate this choice, we compare FL with more computationally intensive methods like K-medoid clustering, which can be viewed as K-means where real data points serve as centroids. As illustrated in Figure 4, the choice of ICL embedding plays a far more critical role in determining performance than the choice of clustering algorithm. When using the same embedding, both FL and K-medoids produce comparable results; however, FL offers a clear advantage in computational efficiency. This makes FL the more practical option for large-scale scenarios without sacrificing performance.

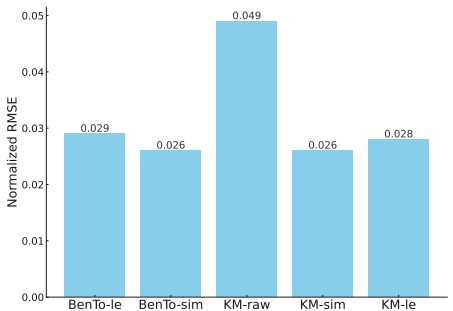

Figure 4: Ablation study on facility location (FL) vs. K-medoids: we report the best NRMSE (lower is better) achieved by each method on MMLU. KM denotes K-medoids. KM-raw, KM-sim and KM-le denote K-medoids on the raw feature matrix $A$, similarity matrix $S$ and $S'$ respectively.

**Choice of LLM.** Our primary experiments employ Llama-2-13B to estimate in-context transferability(ICT) on MMLU. To evaluate whether model scale critically impacts this estimation, we conduct a parallel analysis using the 2.7B-parameter Phi-2

model (Javaheripi et al., 2023). Results show remarkably high alignment between the two models: Spearman correlation reaches $0.9304 \pm 0.0004$ and Pearson correlation $0.9282 \pm 0.0004$. This strong agreement suggests that substantially smaller LLMs can still reliably approximate ICT while offering computational efficiency.

## 6 CONCLUSION

In this study, we demonstrated that large language model (LLM) evaluations can be efficiently conducted with significantly reduced benchmarks, without substantially compromising evaluative accuracy. Utilizing in-context learning to estimate task transferability, our method allows for a reduction of up to 95% in the number of tasks, maintaining less than a 4% deviation from full benchmark results. This approach not only reduces computational and operational costs but also presents a scalable model for rapid LLM assessment. Future work may explore expanding this methodology across different model types and broader task sets to enhance its robustness and applicability in real-world scenarios.

## 7 LIMITATIONS

In this paper, we have focused on achieving cost-efficient benchmark reduction for evaluating large language models (LLMs), which we have demonstrated to be effective through extensive experimentation. However, a notable limitation of this approach is that a smaller benchmark may inherently be less diverse and potentially more vulnerable to adversarial attacks. We recognize that this limitation represents a fundamental trade-off between the efficiency of the evaluation process and the comprehensiveness of the metrics employed.

## ACKNOWLEDGEMENT

The authors gratefully acknowledge the generous gift fund provided by Adobe, which supported this research endeavor. The authors would like to express their sincere appreciation to the reviewers and area chairs for their valuable feedback and insightful comments that significantly contributed to the improvement of this paper.

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

## A  ALGORITHM

Our method BENTO is described in Algorithm 1.

---
**Algorithm 1** Benchmark Task Reduction (BENTO)

---
1: **procedure** TRANSFERABILITYMATRIX(Task, Model, $L$, $M$) ▷ $L$ and $M$ are hyperparameters
2:     $N = \text{length(Task)}$, $p^{(i)} =$ instruction of Task[i]
3:     **for** $i, j = 1 \rightarrow N$ **do**                                    ▷ Estimate ICT from Task[i] to Task[j]
4:         **for** $m = 1 \rightarrow M$ **do**                              ▷ $M$ random seeds
5:             Set random seed to $m$
6:             Sample $L$ exemplars $e_{1:L}^{(i)}$ from source Task[i]
7:             Sample $n_j$ input-output pairs $e_{1:n_j}^{(j)}$ from target Task[j]
8:             Estimate ICT $A_{i,j}$ using Equation (1)
9:         **end for**
10:         Average $A_{i,j}$ over the $M$ random seeds
11:     **end for**
12:     Normalize the columns of $A$ using Equation (2)
13:     **return** $A$
14: **end procedure**
15: **procedure** SIMILARITYMATRIX($A$, $K$)             ▷ Transferability matrix $A$, hyperparameter $K$
16:     Compute the similarity matrix $S$ using Equation (3)
17:     Compute the Laplacian embedding $A'$ using Equation (4)
18:     Compute the cosine similarity matrix $S'$ from $A'$ using Equation (6)
19:     **return** $S, S'$
20: **end procedure**
21: **procedure** BENCHMARKTASKREDUCTION($S$)                    ▷ $S$ can be replaced by $S'$
22:     Maximize Equation (5) by the greedy algorithm
23:     Return $X^*$
24: **end procedure**

---

## B  EXPERIMENT DETAILS

For our main experiments, we use 4 A100 40G for about 3 days. We use $L = 5$ exemplars and $M = 10$ random seeds. We set $n_j$ to be a large value so that we always evaluate on the whole test set. For our main experiments on FLAN, when we normalize $E$, we also divide it by the standard deviation since the metric we use makes the original one too distorted.

Our implementation is based on Klein et al. (2018).

## C  STANDARD DEVIATIONS OF MAIN RESULTS

The ICL accuracy on MMLU are averaged over 10 random seeds. Since the result is a $57 \times 57$ matrix, it's impossible to list the standard deviation for all entries. The average standard deviation over the $57 \times 57$ matrix is 0.017, the standard deviation of the standard deviation is 0.0065.

For our main results presented in Table 2 and Table 3, we compute the error bar via bootstrapping. We randomly sample $M$ models with replacement and compute NRMSE on the sampled models. This process is repeated 1000 times and we compute the mean and standard deviation of the NRMSE. Results are shown in Table 6 and Table 7.

## D  ICL PROMPTS EXAMPLE

On MMLU, we use the following prompts:

Table 6: Error bar of the NRMSE on MMLU. Computed by bootstrapping.

| k\Method | GPT4 | BM25-le | BM25-sim | BENTO-le | BENTO-sim |
|---|---|---|---|---|---|
| 1 | 0.193±0.017 | 0.062±0.015 | 0.349±0.024 | 0.059±0.010 | 0.130±0.013 |
| 2 | 0.073±0.006 | 0.224±0.025 | 0.177±0.020 | 0.090±0.014 | 0.066±0.010 |
| 3 | 0.100±0.014 | 0.220±0.025 | 0.147±0.017 | 0.031±0.009 | 0.168±0.017 |
| 4 | 0.082±0.012 | 0.217±0.023 | 0.122±0.016 | 0.050±0.007 | 0.113±0.012 |
| 5 | 0.060±0.010 | 0.206±0.022 | 0.125±0.015 | 0.045±0.006 | 0.097±0.010 |
| 6 | 0.041±0.005 | 0.193±0.021 | 0.089±0.009 | 0.031±0.006 | 0.042±0.006 |
| 7 | 0.152±0.015 | 0.198±0.021 | 0.157±0.014 | 0.058±0.008 | 0.077±0.007 |
| 8 | 0.168±0.015 | 0.209±0.020 | 0.132±0.011 | 0.159±0.015 | 0.043±0.007 |
| 9 | 0.157±0.014 | 0.205±0.019 | 0.129±0.011 | 0.142±0.013 | 0.033±0.004 |
| 10 | 0.126±0.013 | 0.174±0.016 | 0.113±0.010 | 0.111±0.010 | 0.029±0.005 |

Table 7: Error bar of the NRMSE on FLAN. Computed by bootstrapping

| k\Method | GPT4 | BM25-le | BM25-sim | BENTO-le | BENTO-sim |
|---|---|---|---|---|---|
| 1 | 6.867±3.020 | 0.909±0.400 | 7.276±3.199 | 0.502±0.220 | 0.851±0.374 |
| 2 | 3.021±1.328 | 0.694±0.305 | 3.999±1.758 | 0.427±0.188 | 0.702±0.308 |
| 3 | 1.712±0.752 | 0.559±0.246 | 2.360±1.038 | 0.202±0.089 | 0.033±0.015 |
| 4 | 1.247±0.548 | 0.526±0.232 | 1.552±0.682 | 0.067±0.030 | 0.091±0.041 |
| 5 | 0.814±0.358 | 0.531±0.234 | 1.193±0.524 | 1.321±0.581 | 0.151±0.067 |
| 6 | 0.532±0.234 | 0.594±0.260 | 0.850±0.374 | 0.948±0.417 | 0.272±0.120 |
| 7 | 0.559±0.246 | 0.473±0.208 | 0.598±0.263 | 0.791±0.348 | 0.243±0.107 |
| 8 | 0.377±0.166 | 1.479±0.650 | 0.623±0.274 | 0.582±0.255 | 0.231±0.102 |
| 9 | 0.238±0.105 | 1.223±0.538 | 0.454±0.200 | 0.537±0.236 | 0.307±0.135 |
| 10 | 0.123±0.054 | 1.009±0.444 | 0.329±0.144 | 0.455±0.200 | 0.323±0.142 |
| 11 | 0.087±0.038 | 1.248±0.548 | 0.282±0.124 | 0.363±0.160 | 0.161±0.071 |
| 12 | 0.401±0.176 | 1.249±0.549 | 0.217±0.096 | 0.265±0.117 | 0.191±0.084 |

```
The following are multiple choice questions (with answers)
about [Task A's subject].\n\n[Task A's exemplars][Task B's
question]\nAnswer:
```

An exemplar has the format: `[Question]\nAnswer:  [Answer]\n\n`

On FLAN, we use the following prompts:

```
You are a helpful AI assistant.  Here are some example
input-output pairs that you should follow.\n\n[Task A's
exemplars]Input:\n[Task B's question]\nOutput:
```

An exemplar has the format: `Input:\n[Question]\nOutput:  [Answer]\n\n`

## E    DETAILED PERFORMANCE OF EACH MODEL

The detailed performance of each model is shown in Table 8. Our reduced benchmark works consistently well across different models.

## F    RESULTS ON ADDITIONAL BENCHMARKS

AGIEval (Zhong et al., 2023) is a question-answering dataset where the questions mostly come from real human exams. The questions have diverse sources and forms, but are all formatted as multiple-choice questions. We use the 9 English tasks in this dataset and use ACC as the initial transferability measure. The result is shown in Table 9. This benchmark is not ideal for our setting since the number of tasks is too small, but our method still works relatively well comparing to the baselines. Big-Bench Hard (Suzgun et al., 2022) is a benchmark with 27 subtasks, including

Table 8: Performance of different models on all tasks and selected tasks of MMLU.

| Model | All Tasks | Selected Tasks |
|---|---|---|
| Gemma-7b | 65.2±0.2 | 63.4±0.5 |
| Llama-2-13b | 54.5±0.2 | 53.9±0.2 |
| Llama-2-7b | 46.0±0.1 | 49.8±0.3 |
| Llama-3-8b | 61.7±0.2 | 60.2±1.8 |
| Mistral-7b-v0.3 | 62.1±0.2 | 62.2±0.4 |
| Phi-2 | 56.5±0.3 | 56.7±0.3 |
| Phi-3-mini-4k | 69.5±0.1 | 70.0±0.6 |
| StableLM-2-1.6B | 34.6±0.2 | 34.7±0.7 |
| TinyLlama | 24.9±0.4 | 25.9±1.6 |

Table 9: NRMSE on AGIEval English (lower the better). $k$ is the number of selected tasks. Each number is averaged over different models.

| Method | Best | k=1 | k=2 | k=3 | k=4 | k=5 | k=6 |
|---|---|---|---|---|---|---|---|
| Random | 0.34 | 0.34 | 1.07 | 2.08 | 3.07 | 4.10 | 5.12 |
| GPT4 | 0.07 | 0.48 | 0.32 | 0.20 | 0.16 | 0.15 | 0.07 |
| BM25-le | 0.06 | **0.17** | 0.24 | **0.06** | 0.08 | 0.11 | 0.15 |
| BM25-sim | 0.15 | 0.38 | 0.24 | 0.28 | 0.38 | 0.21 | 0.15 |
| BENTO-le | **0.03** | 0.38 | **0.08** | 0.07 | **0.05** | 0.06 | **0.03** |
| BENTO-sim | **0.03** | 0.53 | 0.23 | 0.20 | 0.16 | **0.03** | 0.08 |

filling-in-the-blank tasks that require a certain natural language response. We require a strict match when computing ACC on this dataset. The result is shown in Table 10. On this dataset, we only use the 3 to 5 examples in the training set as the exemplars. Under this extreme few-shot setting, BM25 seems to work slightly better than our methods.

Table 10: NRMSE on Big Bench Hard (lower the better). $k$ is the number of selected tasks. Each number is averaged over different models.

| Method | Best | k=1 | k=2 | k=3 | k=4 | k=5 | k=6 | k=7 | k=8 |
|---|---|---|---|---|---|---|---|---|---|
| Random | 0.389 | 0.389 | 1.063 | 2.029 | 3.010 | 4.012 | 5.002 | 6.004 | 7.020 |
| GPT4 | 0.072 | 0.540 | 0.174 | **0.086** | 0.099 | 0.110 | 0.212 | 0.138 | 0.072 |
| BM25-le | 0.092 | **0.092** | 0.353 | 0.133 | 0.114 | 0.171 | 0.129 | 0.116 | 0.135 |
| BM25-sim | **0.032** | 0.748 | 0.325 | 0.313 | **0.087** | 0.186 | 0.119 | **0.032** | **0.049** |
| BENTO-le | 0.045 | 0.248 | **0.080** | 0.090 | 0.148 | **0.057** | **0.045** | 0.070 | 0.080 |
| BENTO-sim | 0.154 | 0.780 | 0.392 | 0.333 | 0.388 | 0.179 | 0.154 | 0.196 | 0.219 |

## G  PUBLICLY REPORTED RESULTS ON MMLU

In Table 11, we annotate the publicly reported results for models listed in Table 1. We do not report our assessment of Gemma-7b on BBH because it fails to generate answer in the given format.

Table 11: Comparison of full benchmark performance measured by us and that reported in previous works. The difference may come from different prompts / quantization. BENTO-estimated performance is also attached for comparison.

| Model | Measured Acc(100%) | Reported Acc(100%) | BENTO-estimated Acc(5%) |
|---|---|---|---|
| Llama-2-70b | 66.6 | 68.9 | 67.9 |
| Llama-2-13b | 54.5 | 54.8 | 53.9 |
| Llama-2-7b | 46.0 | 45.3 | 49.8 |
| Llama-3-8b | 61.7 | 69.4 | 60.2 |
| Mistral-7b-v0.3 | 62.1 | 61.1 | 62.2 |
| Phi-2 | 56.5 | 56.7 | 56.7 |
| Phi-3-mini-4k | 69.5 | 70.9 | 70.0 |
| StableLM-2-1.6B | 34.6 | 38.9 | 34.7 |
| TinyLlama | 24.9 | 26.6 | 25.9 |
| Gemma-7b | 65.2 | 64.3 | 63.4 |
| QWEN2.5-7b | 74.2 | 74.2 | 75.8 |
| QWEN2.5-14b | 79.8 | 79.7 | 81.6 |
| GPT-4o-mini | 74.3 | 82.0 | 75.1 |
| GPT-4o | 68.9 | 88.7 | 69.8 |

