# OpenReview forum: "BenTo: Benchmark Reduction with In-Context Transferability"
_ICLR.cc/2025/Conference — ICLR 2025 Poster_

### Official Review · Reviewer_pNg5 · 2024-10-28

**Soundness:** 3
**Presentation:** 3
**Contribution:** 2
**Rating:** 6
**Confidence:** 4

**Summary:**

This paper introduces BENTO, a method designed to efficiently minimize the number of tasks in large language model (LLM) benchmarks while preserving evaluation quality. The core ideas include:
1. In-Context Transferability (ICT): A training-free approach to estimate transferability between task pairs in an LLM benchmark using in-context learning.
2. BENTO: This method formulates task selection as a facility location problem, utilizing task similarities derived from ICT. BENTO identifies the most representative subset of tasks to approximate the evaluation of the complete benchmark.
3. Experiments: Compared to the random selection baseline, BENTO achieves higher evaluation accuracy. It can reduce MMLU and FLAN benchmarks to  5% of the original task, enhancing the LLM evaluation efficiency.

**Strengths:**

1. The paper addresses an intriguing research question concerning the redundancy of data in benchmarks.
2. The use of in-context learning to estimate task transferability is effective.
3. The paper is well-written and easy to comprehend.

**Weaknesses:**

1. Potential Overfitting: Evaluating with a smaller sample size poses a risk of overfitting. It is challenging to ensure that the evaluation data in the benchmark is not used in supervised fine-tuning. Reducing the scale of evaluation data likely increases the chances of data contamination bias. The paper does not simulate or analyze this scenario.
2. Greater Significance for SFT data: This method may be more beneficial for deduplicating training data than for selecting benchmark data. Training typically involves multiple iterations and back-propagation, which entail higher costs and thus require deduplication.
3. Practicality of BENTO for Evaluation Data Reduction: I do not think using BENTO to remove evaluation data as a practical approach. Instead, I view BENTO as a metric for assessing benchmark quality rather than eliminating 95% of existing benchmark data.

**Questions:**

See Weaknesses.

---

> ### Author Response · Authors · 2024-11-21
>
> Thank you for your insightful comments! Below, we address each of your concerns in detail.
>
> > Q1: "Evaluating with a smaller sample size poses a risk of overfitting."
>
> A1: Please refer to Q2/A2 in the general response.
>
> > Q2: "It is challenging to ensure that the evaluation data in the benchmark is not used in supervised fine-tuning. Reducing the scale of evaluation data likely increases the chances of data contamination bias."
>
> A2: Please refer to Q1/A1 in the general response.
>
> > Q3: "This method may be more beneficial for deduplicating training data than for selecting benchmark data. Training typically involves multiple iterations and back-propagation, which entail higher costs and thus require deduplication."
>
> A3: This is an excellent extension of BenTo to training task reduction! In fact, we demonstrate this use case in E2 in the general response, where BenTo is applied to select training tasks for specific target tasks. By identifying high-transferability tasks, BenTo helps focus training on the most relevant subsets of data, reducing computational overhead and increasing efficiency. Your observation aligns well with the broader applicability of our method.
>
> > Q4: "I do not think using BENTO to remove evaluation data as a practical approach. Instead, I view BENTO as a metric for assessing benchmark quality rather than eliminating 95% of existing benchmark data."
>
> A4: Thank you for this perspective! While the primary purpose of BenTo is to provide a lightweight and efficient benchmark for rapid iteration and development of models, it might be necessary to still evaluate the final version of a model on the full benchmark.
>
> Your suggestion to use BenTo as a metric for assessing benchmark quality is particularly interesting. A high-quality benchmark should indeed be harder to reduce effectively, as it would require retaining a larger portion of its data to maintain performance. This aligns with our framework, and we appreciate this novel perspective on leveraging BenTo in this context.

---

> ### Author Response · Authors · 2024-11-24
> **Follow-up Message**
>
> Dear reviewer,
>
> Thank you again for your thoughtful comments! We hope that our response and the updated experimental results have addressed the concerns you raised and provided additional clarity on the key points.
>
> As the discussion period draws to a close, we wanted to kindly check if you had any further questions or aspects you’d like to discuss. Additionally, we were wondering if our clarifications and results have sufficiently addressed your concerns to potentially reconsider your evaluation of the paper.
>
> We greatly appreciate your time and feedback and look forward to hearing any additional thoughts you might have.

---

> > ### Comment · Reviewer_pNg5 · 2024-11-26
> >
> > Thank you for the clarifications. However, I am still concerned that having fewer test samples increases the possibility of the test data being included in the training data. Additionally,  testing these benchmarks may not constitute a significant computational overhead. I would prefer to apply this method during training, and I will maintain my score.

---

> ### Author Response · Authors · 2024-11-26
>
> Dear Reviewer pNg5,
>
> Thanks for your response! However, with all due respect, we would like to argue that our previous response has addressed your concerns with detailed clarifications. To be more specific:
>
> > **fewer test samples increases the possibility of the test data being included in the training data**
>
> Data contamination is out of the main scope of our paper as we focus on improving inference efficiency rather than training. Data contamination is an important open problem and a risk for almost all existing benchmarks because they are public and current large models can easily learn them. **Reducing the size of benchmarks may not make it better, but it does not make it worse**. Instead of growing the benchmark size, keeping the benchmark dynamically changing and non-public is more effective and efficient to resolve this issue. But this is outside the scope of our work.
>
> > **testing these benchmarks may not constitute a significant computational overhead**
>
> We respectfully disagree. It is a widely known fact that the sequential decoding of LLMs in generating responses does suffer from important computational overhead, especially for tasks requiring longer responses. The overhead multiplicatively or exponentially grows because the evaluation of the benchmarks needs to be repeated many times: they have been repeatedly used worldwide to evaluate different models or checkpoints in the development of every single model. In addition, various recent works [1,2,3] also find benchmark reduction important and share the motivations with us.
>
> > **I would prefer to apply this method during training**
>
> We already provided solid experimental results for training task reduction using BenTo in General Response Part2 - E2 BenTo for selecting training data, which demonstrated the effectiveness of our method in improving the training.
>
> We are eager to hear your feedback and opinions on these. Thanks!
>
> ---
> [1] Tinybenchmarks: Evaluating LLMs with fewer examples. 2024
>
> [2] Anchor points: Benchmarking models with much fewer examples. 2024
>
> [3] Improving Model Evaluation using SMART Filtering of Benchmark Datasets. 2024

---

> > ### Author Response · Authors · 2024-12-02
> >
> > Dear Reviewer pNg5,
> >
> > Thanks for your follow-up response to the rebuttal! Your comments and suggestions will greatly help us improve the draft. This is the last day of discussion and we are eager to learn from you whether our latest response further resolved any of your remaining concerns. Can we kindly ask you to consider raising your ratings if our rebuttal and response indeed addressed some of your major concerns (for example, we did report the training results you prefer to see)?
> >
> > Best regards,
> >
> > Authors

---

> > > ### Comment · Reviewer_pNg5 · 2024-12-03
> > >
> > > Thanks for your response. I've raised my score to 6. However, I am very uncertain whether the community will use small-scale data for evaluations in the future and the potential risks this might bring.

---

### Official Review · Reviewer_CaUX · 2024-11-03

**Soundness:** 2
**Presentation:** 3
**Contribution:** 3
**Rating:** 6
**Confidence:** 3

**Summary:**

This paper introduces BENTO for reducing LLMs evaluation benchmarks by selecting representative tasks based on a new metric, In-Context Transferability (ICT), which estimates task similarity without additional training, reducing the computational cost.

**Strengths:**

1.  The BENTO method effectively reduces the number of benchmark tasks to approximately 5% of the original set with minimal impact (<4%) on evaluation accuracy, making LLM evaluation more cost-effective.
2. The proposed ICT metric enables efficient estimation of task similarity without requiring model training or fine-tuning, which lowers resource demands.
3. BENTO demonstrates reliable performance across various LLM benchmarks, indicating its potential applicability to diverse datasets and tasks.

**Weaknesses:**

1. Currently, LLM benchmarks are highly vulnerable to contamination, which diminishes their reliability and credibility. The proposed BENTO method could exacerbate this issue, potentially causing LLMs to focus on a narrower range of tasks, thereby increasing the risk of contamination.
2. It would be valuable to see results on more advanced models, such as Gemini or GPT-4o, to assess the generalizability and scalability of the approach.
3. ICL results can be quite variable. It would be informative to evaluate ICL performance across different models and model sizes.
4. Since ICL requires additional sampling iterations, it would be helpful to provide the associated computation costs for reference.

**Questions:**

N.A

---

> ### Author Response · Authors · 2024-11-21
>
> Thank you for your comment! Our detailed responses regarding each of your concern are listed below.
>
> > Q1: "Currently, LLM benchmarks are highly vulnerable to contamination, which diminishes their reliability and credibility. The proposed BENTO method could exacerbate this issue, potentially causing LLMs to focus on a narrower range of tasks, thereby increasing the risk of contamination."
>
> A1: Please refer to Q1/A1 in the general response.
>
> > Q2: "It would be valuable to see results on more advanced models, such as Gemini or GPT-4o, to assess the generalizability and scalability of the approach."
>
> A2: Thank you for the suggestion. We have included GPT-4o-mini and GPT-4o in our evaluation. Please refer to E3 in the general response.
>
> > Q3: "ICL results can be quite variable. It would be informative to evaluate ICL performance across different models and model sizes."
>
> A3: We have addressed this concern in E1 in the general response. Specifically, we conducted ICL experiments using Phi-2, a smaller model with 2.7B parameters. The results are highly consistent with those obtained using Llama2-13B, the primary model discussed in the paper. This consistency suggests that our method is robust to variations in model size and architecture.
>
> > Q4: "Since ICL requires additional sampling iterations, it would be helpful to provide the associated computation costs for reference."
>
> A4: The computational cost of ICL is given by $\sum_{j=1}^N n_jME$, where:
>
> * $N$ is the number of tasks,
> * $M$ is the number of ICL exemplar sets,
> * $n_j$ is the number of test batches used for the $j$th task, and
> * $E$ is the cost of a single forward pass.
>
> In our experiments:
> * $N \approx 50$,
> * $M = 10$,
> * $n_j \approx 10$ to $20$.
>
> This results in approximately 5000 to 10,000 forward passes. However, this does not weaken the efficiency of our method: please see Q3/A3 in the general response.

---

> > ### Comment · Reviewer_CaUX · 2024-11-22
> >
> > Thanks for your reply. Please include the adjustments in the updated version. I've updated the scores.

---

> > > ### Author Response · Authors · 2024-11-24
> > >
> > > Thank you for your reply! We will surely include all the adjustments in the updated version.

---

### Official Review · Reviewer_TgKk · 2024-11-03

**Soundness:** 3
**Presentation:** 3
**Contribution:** 2
**Rating:** 6
**Confidence:** 4

**Summary:**

The paper presents a novel approach for improving the efficiency of benchmark evaluation in machine learning. The authors propose a method for reducing the number of tasks needed for comprehensive benchmarking by leveraging in-context transferability. This involves identifying and selecting a subset of tasks that can effectively represent a broader set of benchmarks while minimizing redundancy.
(1)	Analyzing the transferability between tasks from the perspective of in-context learning
(2)	The experiments on MMLU and FLAN show that the method proposed by the author can approximate the full performance with 5% of the data.

**Strengths:**

-	Analyzing task transferability from the perspective of ICL presents a novel approach.
-	The method does not require fine-tuning using existing evaluation data; it only needs to conduct ICL transferability tests on some models, which enhances the scalability of the approach.
-	The experimental results demonstrate greater effectiveness compared to selection methods such as random sampling or BM25.

**Weaknesses:**

-	The method raises concerns about its computational efficiency, particularly regarding the actual sizes of the in-context learning (ICL) test samples. For the ICL transfer tests from task i to task j, what are the sizes of the test samples n_j?

Given that the transfer tests from i to j involve N^2 tests (where N is the total number of tasks), the total number of test samples conducted amounts to N^2 \times n_j \times M. Taking MMLU as an example, with N=57, if n_j \times M is only 50 (considering M=10 in appendix B), the total number of tests already reaches 160k, which extremely exceeds the complete MMLU dataset of 12k. Extensive ICL testing suggests that the model has been evaluated on a full dataset, which contradicts the authors' motivation for efficiently reducing the test set size.

-	The method heavily relies on the careful selection of the parameter k. In real scenarios without prior knowledge, determining the appropriate value for k is a significant challenge that could limit the method's usability in practical applications.

-	The approach is currently limited to task-level improvements and does not address example-level enhancements. This limitation may restrict the method's applicability across existing benchmarks, particularly in cases where there are a small number of categories but a large number of examples, such as MATH. Also, the subtasks of dataset FLAN used by the authors, such as ReCoRD and SQuADv2, where the number of examples exceeds 10k, does this indicate a lack of usability for the method?

-	The authors only used 9 models for evaluation, raising concerns about whether the method would still perform consistently across more models. For example, LLaMA2-72B, Qwen2-7B, Qwen2-13B, etc.

**Questions:**

Q1: Will the difference in ICL capabilities of different models lead to bias in the evaluation?
The authors utilize two models (LLaMA 7 b and 13b). However, it is unclear whether their ICL performance is consistent across these models. Additionally, the methodology for integrating data from these models is not specified—do you simply average the results?

Q2: In the author's experimental setup for FLAN's 66 tasks, 100 questions were selected for each task to be considered as "whole benchmark"(Line 274-275). Does this simplification ensure the validity of the experimental results since FLAN contains 700k+ samples?

Q3: The NRMSE is used to compare the performance differences among methods. If the values were to be expressed in terms of absolute accuracy, what would the corresponding values for the other methods be? (Similar to the data presented in Table 1)

Typo: The letter "Q" in Appendix B experiment details has not been defined.

---

> ### Author Response · Authors · 2024-11-21
> **Part1**
>
> Thank you for your detailed and thoughtful comments! We have addressed each of your concerns below.
>
> >Q1: "For the ICL transfer tests from task i to task j, what are the sizes of the test samples n_j?"
>
> A1: In our experiments, we use approximately 100 test samples for most tasks. Batch decoding with a batch size of 8 leads to 10–20 forward passes for each ICL exemplar set. Since we have $M=10$ exemplar sets, the total number of forward passes required for each task is 100–200. Note this only need to be done once. Then the reduced benchmark can be repeatedly to replace the original benchmark in many scenarios, e.g., evaluating other models, hyperparameter tuning, etc.
>
> >Q2: "...Extensive ICL testing suggests that the model has been evaluated on a full dataset, which contradicts the authors' motivation for efficiently reducing the test set size."
>
> A2: Please refer to Q3/A3 in the general response.
>
> >Q3: "The method heavily relies on the careful selection of the parameter k. In real scenarios without prior knowledge, determining the appropriate value for k is a significant challenge that could limit the method's usability in practical applications."
>
> A3: The nature of the greedy algorithm for facility location allows for an adaptive and automatic selection of $k$ with minimal prior knowledge. Specifically, the facility location is submodular (diminishing return) so the objective value will saturate as more tasks being added by the greedy procedure. Hence, we can determine an adaptive $k$ by trying a much larger $K$ and find $1\leq k\leq K$ where the saturation happens or when the transferability is sufficiently large. We will investigate it in more detail for future works.
>
> >Q4: "The approach is currently limited to task-level improvements and does not address example-level enhancements."
>
> A4: As discussed in the related works section and section 5.2's experiments ("Task selection and example selection" paragraph), our method can be seamlessly combined with any example-level reduction methods and further improve them. Therefore, we consider task-level and example-level benchmark reduction as two parallel directions.
>
> >Q5: "...the subtasks of dataset FLAN used by the authors, such as ReCoRD and SQuADv2, where the number of examples exceeds 10k, does this indicate a lack of usability for the method?"
>
> A5: Our method does not require evaluation on the full test set. As outlined in E2 in the general response, we conduct ICL testing on only 50 samples from the validation set while evaluating on the full test set. Despite this reduced samples for ICT estimation, our method still achieves promising performance, demonstrating its usability and efficiency even for large-scale datasets.

---

> ### Author Response · Authors · 2024-11-21
> **Part2**
>
> >Q6: "The authors only used 9 models for evaluation, raising concerns about whether the method would still perform consistently across more models. For example, LLaMA2-72B, Qwen2-7B, Qwen2-13B, etc."
>
> A6: We appreciate this suggestion and have added evaluations for additional models, including LLaMA2-70B, Qwen2.5-7B, Qwen2.5-14B, GPT-4o-mini and GPT-4o. Please refer to E3 in the general response.
>
> >Q7: "Will the difference in ICL capabilities of different models lead to bias in the evaluation?"
>
> A7: As demonstrated in E1 in the general response, we used Phi-2, a 2.7B parameter model, for ICL testing. The results showed high correlation with those obtained using Llama2-13B, indicating that our method is robust to variations in ICL capabilities across models. This consistency minimizes the risk of evaluation bias.
>
> >Q8: "...the methodology for integrating data from these models is not specified—do you simply average the results?"
>
> A8: Apologies for the confusion. We only use one model for ICL testing and estimate ICT only once on each benchmark: as mentioned in Section 5, "ICL is performed on Llama-2-13B and Llama-2-7B to estimate ICT for MMLU tasks and FLAN tasks, _respectively_." We used the 7B model for FLAN as our computational resources were limited for evaluating some particularly long examples with the 13B model.
>
> >Q9: "In the author's experimental setup for FLAN's 66 tasks, 100 questions were selected for each task to be considered as "whole benchmark"(Line 274-275). Does this simplification ensure the validity of the experimental results since FLAN contains 700k+ samples?"
>
> A9: The subsampled FLAN is used as a new evaluation benchmark that is not necessarily of the same distribution as the original FLAN. We demonstrate BenTo's effectiveness and efficiency on further reducing this benchmark—featuring diverse task formats.
>
> When FLAN is used for training, we showed that our method is applicable to the full dataset of FLAN tasks; please refer to E2 in the general response for details.
>
> > Q10: "The NRMSE is used to compare the performance differences among methods. If the values were to be expressed in terms of absolute accuracy, what would the corresponding values for the other methods be? (Similar to the data presented in Table 1)"
>
> A10: Please refer to the E3 in the general response.
>
> > Q11: "The letter "Q" in Appendix B experiment details has not been defined."
>
> A11: Apologies for the oversight. $Q$ denotes the sizes of the test samples.

---

> ### Author Response · Authors · 2024-11-24
> **Follow-up Message**
>
> Dear reviewer,
>
> Thank you again for your thoughtful comments! We hope that our response and the updated experimental results have addressed the concerns you raised and provided additional clarity on the key points.
>
> As the discussion period draws to a close, we wanted to kindly check if you had any further questions or aspects you’d like to discuss. Additionally, we were wondering if our clarifications and results have sufficiently addressed your concerns to potentially reconsider your evaluation of the paper.
>
> We greatly appreciate your time and feedback and look forward to hearing any additional thoughts you might have.

---

### Official Review · Reviewer_3D6d · 2024-11-08

**Soundness:** 3
**Presentation:** 3
**Contribution:** 3
**Rating:** 6
**Confidence:** 3

**Summary:**

This paper addresses the challenge of evaluating large language models (LLMs) efficiently by proposing a method to reduce the size of LLM benchmarks without compromising the quality of evaluation. Traditional benchmarking approaches are costly and resource-intensive due to the large number of tasks included to cover various LLM capabilities. The authors introduce In-Context Transferability (ICT) as a training-free method to estimate task transferability by leveraging in-context learning, allowing them to assess which tasks are likely to improve performance on others. Using ICT, they propose Benchmark Task Reduction (BENTO), which formulates the task selection as a facility location problem to select a representative subset of tasks. Experiments show that BENTO can reduce benchmark tasks by up to 95% while maintaining consistent evaluation results, offering a cost-effective solution for LLM benchmarking.

**Strengths:**

1.By reducing the benchmark to just 5% of its original size without sacrificing evaluation quality, BENTO significantly lowers the computational and financial burden of LLM evaluation.

2.The paper’s novel use of in-context learning for transferability estimation provides a scalable, training-free solution for task reduction, making it more practical than traditional methods that rely on finetuning.

3.The ICT and BENTO methods can be applied to various LLM research and evaluation scenarios, potentially benefiting a wide range of applications beyond just benchmarking.

**Weaknesses:**

1.The reliance on in-context learning may limit the generalizability of the approach, as it assumes that in-context performance accurately reflects task transferability, which may not hold true across all tasks or models, especially for weak LLMs. More experiments for weak LLMs should be added.

2.Although the paper demonstrates consistency with reduced benchmarks on certain datasets, additional testing across more diverse tasks and benchmarks could strengthen the validity of the results. Reducing a benchmark to a small subset of tasks may overlook nuanced skills or specific task requirements that LLMs need to perform well in specialized applications, potentially leading to incomplete evaluations in some cases.

3.It is necessary to analyze if the reduced benchmark would cause the hurt of robustness for benchmark leakage issue or overfitting to few dimensions.

**Questions:**

Please refer to weaknesses

---

> ### Author Response · Authors · 2024-11-21
>
> Thank you for your thoughtful comments! Below, we provide detailed responses to each of your concerns.
>
> >Q1: “... it assumes that in-context performance accurately reflects task transferability, which may not hold true across all tasks or models, especially for weak LLMs. More experiments for weak LLMs should be added.”
>
> A1: We verified the assumption that in-context transfer learning's performance reliably reflects task transferability. To further demonstrate the assumption, we have provided additional experiments with Phi-2, a weaker LLM with only 2.7B parameters much smaller than the Llama2-13B in the main paper. The results, as outlined in E1 in the general response, show high consistency between the transferability estimated using Phi-2 and that of Llama2-13B. These findings strongly imply that the assumption is solid and holds even on weaker models, validating our approach across varying LLM capacities.
>
> >Q2: "...Reducing a benchmark to a small subset of tasks may overlook nuanced skills or specific task requirements that LLMs need to perform well in specialized applications, potentially leading to incomplete evaluations in some cases."
>
> A2: As discussed in the limitations section of the paper, there may exist an inherent trade-off between evaluation efficiency and comprehensiveness (the number of evaluation tasks). The very aim of BenTo is to stay on the Pareto frontier, and, **maximally preserve the evaluation results on all the benchmark tasks by only evaluating models on a given budget of tasks**. While a few nuanced and rare skills or specific task requirements may inevitably be overlooked, BenTo preserves most important skills to be evaluated by selecting representative tasks with the greatest transferability to others.
>
> >Q3: "It is necessary to analyze if the reduced benchmark would cause the hurt of robustness for benchmark leakage issue or overfitting to few dimensions."
>
> A3: Please refer to Q1/A1 and Q2/A2 in the general response.

---

> ### Author Response · Authors · 2024-11-24
> **Follow-Up Message**
>
> Dear reviewer,
>
> Thank you again for your thoughtful comments! We hope that our response and the updated experimental results have addressed the concerns you raised and provided additional clarity on the key points.
>
> As the discussion period draws to a close, we wanted to kindly check if you had any further questions or aspects you’d like to discuss. Additionally, we were wondering if our clarifications and results have sufficiently addressed your concerns to potentially reconsider your evaluation of the paper.
>
> We greatly appreciate your time and feedback and look forward to hearing any additional thoughts you might have.

---

> > ### Author Response · Authors · 2024-11-28
> > **Reminder**
> >
> > Dear Reviewer 3D6d,
> >
> > Thank you again for your thoughtful comments! We have carefully responded to your comments with further clarifications above and additional experiments in the general response at the top of this page. We haven't heard from you yet and we are eager to hear your feedback. Thanks!
> >
> > Authors

---

> ### Author Response · Authors · 2024-12-01
>
> Dear Reviewer 3D6d,
>
> Thanks for your positive ratings for the soundness, presentation, and contribution of our paper! We provided new experimental results as requested in your original comments. To summarize:
>
> - We showed that our method is still effective on much smaller LLMs.
> - As previous work [1-3] pointed out, benchmark reduction is significant as the benchmarks are supposed to be used for hundreds or thousands of times in the development of many LLMs. There may exist a trade-off between the saved evaluation cost and the robustness/coverage, but we showed that the saving is big and the negative impact is minimal.
> - Data leakage is an important open problem. Reducing benchmark size may not make it better, but it does not make it worse, because current LLMs can easily fit and memorize the original benchmarks as well. Instead of growing the benchmark size, keeping the benchmark dynamically changing and non-public is more effective and efficient to resolve this issue. But this is outside the main scope of our work.
> - Overfitting to a few dimensions does not happen in our study since our method selects the tasks with the greatest transferability to others, i.e., the ones that bring the greatest improvement to generalization capabilities. We further demonstrated it by new experiments finetuning LLMs on BenTo selected source tasks.
>
> Would you please check if our rebuttal and clarifications above addressed your concerns? We are eager to hear your feedback and would like to answer any further questions before the discussion period ends (which is in two days). Thanks!
>
> Best regards,
>
> Authors
>
> - - -
>
> [1] Tinybenchmarks: Evaluating LLMs with fewer examples. 2024
>
> [2] Anchor points: Benchmarking models with much fewer examples. 2024
>
> [3] Improving Model Evaluation using SMART Filtering of Benchmark Datasets. 2024

---

> > ### Comment · Reviewer_3D6d · 2024-12-03
> > **Thanks for the responses**
> >
> > I raise the score to 6, as the authors have addressed most of my concerns.

---

> > > ### Author Response · Authors · 2024-12-03
> > > **Thanks for the feedback!**
> > >
> > > Dear Reviewer 3D6d,
> > >
> > > Thank you for confirming that your concerns have been addressed! We will keep improving our draft and incorporate all our discussion results into future versions. Thanks again for your support!
> > >
> > > Best regards,
> > >
> > > Authors

---

### Author Response · Authors · 2024-11-21
**General Response Part1: Most commonly asked questions**

Dear reviewers,

Thank you for your insightful and constructive feedback! To address the key concerns raised, we have included additional experimental results (in the next few parts of the general response) and detailed responses to the most commonly asked questions. We appreciate your time and effort in reviewing our work and hope our clarifications provide a comprehensive resolution to your concerns.

> Q1: Will this method increase the risk of data leakage/contamination?

A1: Although benchmark leakage and contamination is out of the main scope of this paper, our method does not exacerbate the issue and may even mitigate it. Specifically, if certain tasks are fully leaked:
- When they are target tasks in the estimation of ICT, we get higher ICL accuracy on them no matter what the source tasks are. But since BenTo normalizes the accuracy for each target task (through zero-centering), the resulted transferability w.r.t. different source tasks will be similar and close to zero. So they will not sincerely affect the task selection objective (which aims to maximize the transferability).
- When they are source tasks (the provider of ICL demos) in the estimation of ICT, they bring limited additional gains to the transferability since the model already memorized them. This lowers the transferability scores of the leaked source tasks. Since BenTo is designed to select source tasks with high transferability, the chance of leaked tasks to be selected is reduced.

> Q2: Will this method cause overfitting?

A2: BenTo is an evaluation benchmark reduction method and it does not lead to overfitting:
- BenTo selects tasks with higher transferability (generalizability) to other tasks. So it removes the redundancy of the benchmark while preserving the diversity and generalizability instead of hurting them.
- BenTo-selected tasks do not lead to overfitting when used for training. Instead, they improves the generalization and saves the training cost. This has been verified by the experiments in our general response: the resulting model achieves better performance on unseen target tasks that do not overlap with the selected source tasks. This demonstrates that our method does not suffer from overfitting.

>Q3: The ICL stage of BenTo requires more forward passes than evaluating a single model on the full benchmark. Does this mean that BenTo is inefficient?

A3:
- While ICL testing requires more computation than evaluating a single model on the benchmark, this cost is incurred only once but the reduced benchmark can be repeatedly used to save computation forever. Please think about how many times each popular benchmark have been repeatedly used worldwide everyday.
- Our method can significantly shorten the development of new models or product by reducing the validation cost per iteration for each version of the model or every configuration of the training.
- Compared to existing approaches of transferability estimation, which depend on model training or gradient/Hessian computation, our ICL-based approach is much more efficient due to its training-free and gradient-free nature.
- Our method's cost can be further reduced by using a smaller model for ICL or using fewer samples. As highlighted in the general response, we can replace Llama-13B with a 2.7B LLM, without inducing significant performance degradation. Moreover, as reported in E2 in the general response, the samples used in ICL can be much fewer than the full test set without affecting the performance.

---

### Author Response · Authors · 2024-11-21
**General Response Part2: Additional Experimental Results E1 and E2**

## E1: Weak Language Model for ICL transferability estimation
We demonstrate that a weaker/smaller model can be used to estimate in-context learning transferability (ICT), consistently yielding promising performance on benchmark task reduction. Specifically, we apply Phi-2, a smaller LLM with only 2.7B parameters (our main paper uses Llama2-13B), to estimate ICT between MMLU tasks, under the same experimental setup as in the main paper. The correlation metrics between the  ICT estimated by Phi-2 and Llama2-13B are:
| Pearson   | Spearman    |
| ---    | ---   |
| 0.9282 ± 0.0004 | 0.9309 ± 0.0004 |

The high linear and rank correlation indicate a high degree of consistency across models of varying sizes on estimating ICT. Notably, BenTo, when applied with Phi-2 estimated ICT, still achieves 4% normalized root mean square error (NRMSE) in the evaluation of LLMs, using approximately 4% of the total tasks in MMLU.

## E2: BenTo for selecting training data
We further showcase the versatility of ICT by applying BenTo to selecting source tasks (i.e., training tasks) tailored to specified target tasks. In this setup, BenTo aims to maximize the transferability of source tasks to a specified group of target tasks.

We categorize FLAN tasks into four groups based on their output format: classification, closed-form generation, open-ended generation, and translation-like questions. For tasks containing multiple formats, we split them into subsets, each focusing on a single format. From each group, we sample three sets of target tasks (with size 3–5 tasks per set) and use BenTo to select the source tasks for each set. We compare this approach against a random baseline that randomly draws the same number of source tasks (evaluated over three random seeds).

Setup Details:

+ ICT is estimated on $\leq$ 50 validation samples per task.
+ Training is conducted on 50% of the training set
+ Evaluation is conducted on the full test set.
+ Evaluation Metrics: Accuracy is reported for classification and closed-form generation tasks, while loss values (i.e., logarithm of perplexity) is used for other tasks.

The results are summarized in the following tables. The last two columns are the test-set performance of the model trained on tasks selected by BenTo and random baseline, respectively.

For the first six tasks evaluated by accuracy, **higher is better**:

| Target tasks   | Category                | BenTo | Random|
|----------------|-------------------------|-------------------|--------------------|
| record, qnli, anli_r2 | classification | **0.507** | 0.325±0.075 |
| cb, cola, sentiment140, story_cloze | classification | **0.799** | 0.7±0.053 |
| snli, imdb_reviews, qnli, sentiment140, wsc | classification | **0.886** | 0.741±0.066 |
| quac, drop, fix_punct | closed-form generation | **0.275** | 0.213±0.095 |
| natural_questions, trivia_qa, squad_v2 | closed-form generation | **0.382** | 0.181±0.01 |
| math_dataset, true_case, squad_v1, coqa | closed-form generation | **0.447** | 0.305±0.06 |

For the last six tasks evaluated by loss, **lower is better**:

| Target tasks   | Category                | BenTo | Random|
|----------------|-------------------------|-------------------|--------------------|
| rte, sst2, wmt16_translate_fien | open-ended generation | **2.91** | 3.17±0.05 |
| common_gen, cola, wmt16_translate_roen, wmt14_enfr | open-ended generation | **2.33** | 2.94±0.1 |
| wmt16_translate_tren, openbookqa, cnn_dailymail, anli_r1, wmt16_translate_csen | open-ended generation | **1.98** | 2.08±0.03 |
| wmt16_translate_csen, opinion_abstracts_rotten_tomatoes, wmt16_translate_tren | translation-like | **1.31** | 1.38±0.02 |
| wmt16_translate_ruen, wmt16_translate_deen, wmt14_enfr | translation-like | **0.9** | 0.91±0.01 |
| para_crawl_enes, ag_news_subset, wmt16_translate_fien | translation-like | **1.35** | 1.51±0.1 |

These results demonstrate that BenTo can be extended to training task selection: it consistently and significantly outperforms the random baseline. Notably, this shows that BenTo can operate effectively even on a large dataset while relying on relatively few ICL samples, highlighting its efficiency and practicality in real-world applications.

---

### Author Response · Authors · 2024-11-21
**General Response Part3: Additional Experimental Results E3**

## E3: Additional Models for Evaluation on MMLU
Following reviewers' suggestions, we add 5 additional models of diverse sizes on MMLU experiments: QWEN2.5-7b, QWEN2.5-14b, Llama2-70b, GPT-4o-mini and GPT-4o. For GPT-4o-mini and GPT-4o, we use the following system prompt: "You are a helpful assistant. Say a single letter that correspond to the correct answer." For fairness, we apply the same user prompt to all models.

The results are reported in the following table, alongside with the results of the major baselines. The last 3 columns report the difference of accuracy to the ground truth (evaluation on all tasks) caused by each task reduction method. BenTo induces the smallest difference.

difference between predicted accuracy and the ground truth full benchmark accuracy.
| Model            | Ground Truth (100%) | BenTo (5%) | ChatGPT (5%) | BM25 (5%) |
|------------------|---------------------|------------|--------------|-----------|
| Llama2-13b       | 54.5                | -0.6       | +5.4         | -6.8      |
| Llama2-7b        | 46.0                | +3.8       | +6.8         | -4.6      |
| Llama2-70b       | 66.6                | +1.3       | +6.2         | -9.0      |
| Llama3-8b        | 61.7                | -1.5       | +4.2         | -11.6     |
| Mistral-7b-v0.3  | 62.1                | +0.1       | +4.9         | -10.6     |
| Phi-2            | 56.5                | +0.2       | +5.1         | -8.2      |
| Phi-3-mini-4k    | 69.5                | +0.5       | +2.1         | -12.2     |
| StableLM-2-1.6B  | 34.6                | +0.1       | -1.6         | -3.5      |
| TinyLlama        | 24.9                | +1.0       | +1.4         | -0.4      |
| Gemma-7b         | 65.2                | -1.8       | +9.0         | -10.8     |
| QWEN2.5-7b       | 74.2                | +1.6       | +4.2         | -14.7     |
| QWEN2.5-14b      | 79.8                | +1.8       | +3.5         | -14.0     |
| GPT-4o-mini      | 74.3                | +0.8       | +5.0         | -11.6     |
| GPT-4o           | 68.9                | +0.9       | -2.2         | -8.8      |

---

### Meta-Review · Area_Chair_3toJ · 2024-12-17

**Metareview:**

In this paper, the authors aimed to reduce the number of benchmark tasks to efficiently evaluate LLMs by measuring the transferability between tasks. They introduced a cost-efficient and training-free approach to measuring the transferability based on ICL.

Overall, the idea is interesting. The proposed approach looks reasonable and demonstrated to be effective with promising experimental results.  Original concerns raised by the reviewers were focused on the clarification of the proposed method and the comprehensiveness of the experiments. Most of the concerns have been addressed during the rebuttal.

Therefore, I recommend acceptance for this paper.

**Additional Comments On Reviewer Discussion:**

Original concerns raised by the reviewers were focused on the clarification of the proposed method and the comprehensiveness of the experiments. Most of the concerns have been addressed during the rebuttal.

---

### Decision · Program_Chairs · 2025-01-22

Accept (Poster)